# Leprosy in skeletons from archaeological sites: A systematic review

Hugo Pessotti Aborghetti[1], Simon M. Collin [1], Julienne Dadalto dos Santos[1], Pamela Barbosa dos Santos[1], Taís Loureiro Zambon[1], Rafael Maffei Loureiro[2], Patrícia D. Deps [1,2,3]*

1 Department of Social Medicine, Federal University of Espírito Santo, Vitória, Espírito Santo, Brazil, 2 Medical Image Bank, Department of Radiology, Hospital Israelita Albert Einstein, São Paulo, São Paulo, Brazil, 3 Postgraduate Programme in Infectious Diseases (PPGDI), Federal University of Espírito Santo, Vitória, Espírito Santo, Brazil

* patricia.deps@ufes.br

## Abstract

### Background

Leprosy (Hansen's disease) is an ancient stigmatising infectious disease that remains endemic in many countries. Leprosy-related bone changes that cause disabilities in affected persons are evident in skeletons from archaeological sites. The aim of our synthesis of paleopathological data was to gain insights into the disease's historical distribution and presentation.

### Methodology

Systematic review of paleopathological studies describing human remains with signs of leprosy published up to December 2023. Extracted data on bone features from skulls and limbs, including rhinomaxillary syndrome (RMS) in cranial bones and post-cranial bone changes (PCBC) in hands and feet, were summarised, together with genomic data from studies of *Mycobacterium leprae* ancient DNA.

### Findings

The 297 skeletons described in 67 studies comprised 264 skeletons from sites in modern-day Europe (117 from England, 68 from Denmark); 23 skeletons from Asia (10 from India), 5 from The Americas, and 4 from the African continent (all from Egypt); 174 (58.6%) were from *leprosaria*, 255 (85.9%) were adults, 28 (9.4%) adolescent, 14 (4.7%) of indeterminate age. Skeletons dated from 3715 BCE to 1839 CE, peaking around the 15th Century. Probable and possible RMS were identified in 85 (30.5%) and 153 (54.8%) of 279 skeletons with cranial data, respectively. Lower limb pathological PCBC were most prevalent in tarsals (76.6%), metatarsals (81.5%), and feet phalanges (85.6%). In upper limbs, 75.8% of humeri, 65.8% of radii, 61.0% of ulnae and 75.8% of

**Data availability statement:** All relevant data are in the manuscript and its supporting information files.

**Funding:** The author(s) received no specific funding for this work.

**Competing interests:** The authors have declared that no competing interests exist.

hand phalanges exhibited pathological alterations. From 73 skeletons from 19 genomic studies, *M. leprae* single nucleotide polymorphism (SNP) type 3 was identified in 59 skeletons (80.8%), SNP type 2 in 11 (15.1%), type 4 in two, and type 1 in one.

## Conclusions

Four out of five archaeological skeletons with leprosy exhibited some degree of RMS, which is pathognomonic of the most severe form of the disease, irrespective of whether the skeleton was excavated from a leprosarium (leprosy hospital) or from a public cemetery or other burial site. The relatively small numbers of remains excavated over a wide geographical area and a long time period, and the focus of archaeological studies on skeletons already identified as having leprosy, mean that it is difficult to prove or disprove theories that aim to explain the decline and eventual disappearance of leprosy as a disease in Europe.

## Author summary

The study of leprosy in human remains from archaeological sites has the potential to improve our understanding of the disease in the present day. Our review presents a summary of bone changes caused by leprosy in ancient skeletons, based on studies conducted mainly in the region of modern-day Europe. We found that changes to the skull corresponding to a severe form of leprosy were very common, affecting 4 out of 5 skeletons, irrespective of whether the skeleton was excavated from a *leprosarium* (leprosy hospital) or from a public cemetery or other burial site. The relatively small numbers of remains excavated over a wide geographical area and a long time period, and the focus of archaeological studies on skeletons already identified as having leprosy, mean that it is difficult to prove or disprove theories that aim to explain the decline and eventual disappearance of leprosy as a disease in Europe. Newer techniques such as those that test for genetic evidence of the bacteria that causes leprosy or that look at microscopic bone changes have the potential to provide information about how the disease progresses in people affected by leprosy in countries where it still occurs today and about how leprosy is transmitted from infected animals to the human population and the role of this 'zoonotic' transmission of the disease in sustaining leprosy as a major public health problem in countries such as Brazil.

## Introduction

Leprosy, known today by the non-stigmatising name of Hansen's disease, is a chronic infectious disease that primarily affects the skin and peripheral nerves. It is caused by the bacilli *Mycobacterium leprae* and *Mycobacterium lepromatosis* with an incubation period that ranges from 3 to 7 years [1]. The Ridley-Jopling clinical immunological classification of leprosy describes two poles [2], ranging from the non-infectious

tuberculoid (TT) paucibacillary (PB) form to the infectious lepromatous (LL) multibacillary (MB) form, the latter being implicated in most transmission of the disease [1]. Leprosy is classified as a Neglected Tropical Disease, with approximately 200,000 new cases diagnosed annually and at least 4 million people living with lifelong physical disabilities [3,4].

Leprosy has been well-explored in paleopathology, with dozens of excavations since the late 1950s [5–9], mostly in northern and western Europe, describing bone alterations in skeletons dating back several millennia [10,11]. Leprosy is an ancient human disease documented in historical texts from various regions and, until the 16th century, it was a significant and widespread health issue in Europe [11–13]. However, the medieval period saw a drastic decline in the incidence of leprosy in European populations [11,12]. The exact reasons for this decline remain undetermined, though several hypotheses exist, suggesting a multifactorial cause. These include the gradual elimination of MB forms of leprosy in favour of PB leprosy [14], protective cross-immunity between the main causative agent of leprosy (*M. leprae*) and *M. tuberculosis* [15], and competing effects on mortality attributed to leprosy and tuberculosis co-infection [16–19]. Another hypothesis links the decline of leprosy to climatic changes, such as the 'Medieval Warm Period' (950–1250 AD) followed by the 'Little Ice Age' (1275–1455 AD) [20–22]. Regardless of the underlying causes, it is generally accepted that the prevalence of leprosy in Europe peaked around the 11th-13th centuries, when many *leprosaria* (lazar houses/hospitals) were established, declined during the 14th-15th centuries, and had mostly disappeared by the 18th century, although a small number of autochthonous cases have occurred in southern Europe in the 21st Century [23,24]. As seen in the modern era, improved living conditions [25–27] and better nutrition [28,29] may have also contributed to this decline.

Bone changes in leprosy are categorized as specific, non-specific, and osteoporotic [30], and subsequent disabilities occur mainly in the hands, feet, and face [31]. Non-specific bone lesions are the most common, resulting in osteoarthropathy in hands and feet [32], whereas specific bone changes, caused by bacillary invasion in facial bones [33], occur in 3–5% of clinical cases [34]. Oral and nasal involvement is observed more frequently in MB leprosy, especially in LL cases. The nasal mucosa, which serves as the primary entry point for infection [35], is invaded by *M. leprae* in 95% of LL cases, whereas invasion of oral mucosa is observed in up to 60% of LL cases [36]. Leprosy-related oral lesions can create an environment conducive to bacterial infection, potentially leading to dental abscesses [37]

Rhinomaxillary syndrome (RMS) was defined by Johs Andersen and Keith Manchester in 1992 based on examination of skulls from medieval archaeological sites in England [38]. Maxillary and nasal bone changes are considered pathognomonic of the LL form of leprosy, in which *M. leprae* infection of the nasal passages and palate leads to the collapse of the nasal bridge, resorption of the central part of the maxilla, inflammation of the floor and walls of the nasal cavity and, ultimately, perforation of the hard palate [8,39]. Andersen and Manchester systematised these changes into seven criteria defining RMS. These alterations manifest as facial profile changes, including a 'saddle' nose, concave middle-third of the face, reduced maxillary projection, and inverted upper lip [40,41].

The primary objective of this systematic review was to characterize leprosy-related bone changes reported in archaeological studies, quantifying as primary outcomes the proportion of human remains in which probable or possible RMS and/or post-cranial alterations could be identified. Our secondary objectives were to describe variation in RMS over time in Europe, to investigate correlations of RMS with post-cranial alterations, to describe the frequency of bone alterations by age group, and to summarize *M. leprae* genotypes identified from ancient DNA (aDNA). Our underlying aim is to use knowledge acquired from paleopathology to better understand the history of leprosy and its impact on societies and affected individuals from ancient times to the present day.

## Results

### Included studies

Searches returned 1,831 unique references (Fig 1 and S1 Appendix), of which 237 were screened in full; 86 were included for analysis, comprising 70 non-genomic studies, of which 3 were rated as 'C' quality and not carried forward for data

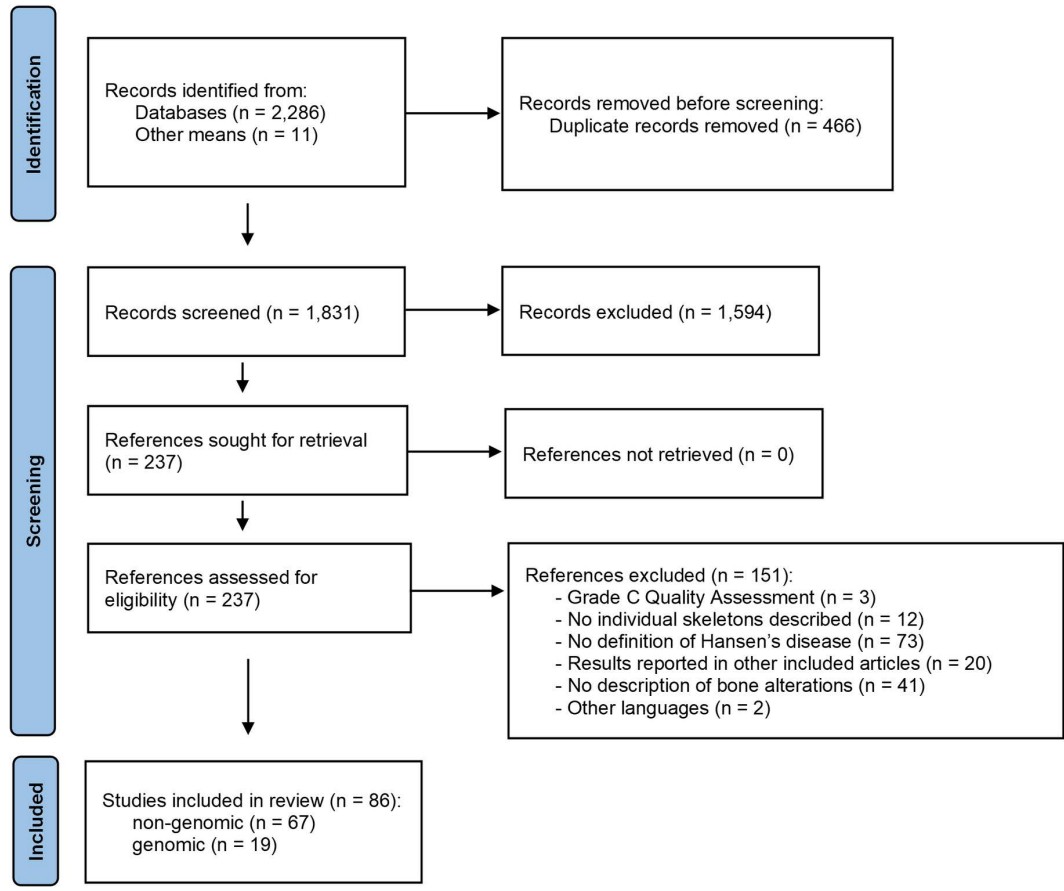

**Fig 1. Flow diagram of studies describing skeletons with leprosy from archaeological sites.**

extraction; 19 references, all rated 'A' or 'B', reported genomic analysis of human remains previously described in non-genomic studies.

## Geographic and demographic data (non-genomic studies)

The 67 non-genomic studies described 297 individual skeletons; 53 studies described 264 skeletons from sites in modern-day Europe, including 1 study in Austria [42], 1 Croatia [43], 1 Cyprus [44], 2 Czech Republic [45,46], 5 Denmark [5–7,47,48], 15 England [49–63], 2 France [64,65], 2 Germany [66,67], 7 Hungary [67–73], 2 Ireland [74,75], 8 Italy [76–83], 1 Norway [84], 3 Portugal [85–87], 1 Slovakia [88], 2 Sweden [89,90]) (Table 1). Sixteen studies were outside Europe: 3 in Armenia [91–93], 1 Easter Island [94], 1 Egypt [95], 1 Georgia [96], 2 India [97,98], 2 Japan [99,100], 1 former Netherlands Antilles [101], 1 Suriname [102], 1 St. Vincent and the Grenadines [103], 1 Thailand [104], 1 Turkey [81], 1 Uzbekistan [105] (Table 1). The 264 skeletons from sites located in modern-day Europe included 117 from England and 68 from Denmark. Studies from sites in Asia described 23 skeletons, of which 10 were from India. Studies from The Americas described 5 skeletons, including 3 in the Netherlands Antilles. The African continent's 4 skeletons were all from Egypt (Table 1).

Of the 297 individual skeletons, 14 (4.7%) had an undefined age at death, 23 (7.7%) were adolescent (10–18 years old based on mid-point of estimated age if no burial record), 5 (1.6%) were <10 years old, and 255 (85.9%) were adults

**Table 1. Geographic and chronological range of skeletons with leprosy by continent.**

| Country | Number of skeletons | Continent | Earliest to most recent age | Reference |
|---|---|---|---|---|
| Egypt | 4 | Africa (4 skeletons) | 2nd century BCE | [95] |
| Netherlands Antilles | 3 | America (5 skeletons) | St. Vincent and the Grenadines: ~1806 CE<br>Suriname: 1866–1896 CE | [103]<br>[102] |
| St. Vincent and the Grenadines | 1 | | | |
| Suriname | 1 | | | |
| Armenia | 3 | Asia (23 skeletons) | India: 2550–2030 BCE<br>Japan: mid18th/ early 19th C | [98]<br>[100] |
| Cyprus | 1 | | | |
| Georgia | 2 | | | |
| India | 10 | | | |
| Japan | 3 | | | |
| Thailand | 2 | | | |
| Turkey | 1 | | | |
| Uzbekistan | 1 | | | |
| Austria | 1 | Europe (264 skeletons) | Hungary: 3780–3650 BCE*<br>England: 1839 CE | [70]<br>[59] |
| Croatia | 2 | | | |
| Czechia | 3 | | | |
| Denmark | 68 | | | |
| England | 117 | | | |
| France | 24 | | | |
| Germany | 3 | | | |
| Hungary | 11 | | | |
| Ireland | 4 | | | |
| Italy | 13 | | | |
| Norden Ireland | 1 | | | |
| Norway | 1 | | | |
| Portugal | 8 | | | |
| Slovakia | 1 | | | |
| Sweden | 7 | | | |
| Easter Island | 1 | Oceania (1 skeleton) | Late 19th/ early 20th CE | [94] |

Legend. In the table, the asterisk (*) denotes the earliest reported case of leprosy across all geographic regions and time periods; BCE = Before Common Era; CE = Common Era.

(19+years old); 46 (15.5%) had undefined sex, 81 (27.3%) were female, and 170 (57.2%) were male (Table 2). The median age of the 23 adolescents was 15.5 years; the 5 children comprised one infant (age 4–5 months) [106], three children aged 4–5 years and one aged 8–9 years old. *Leprosaria* contributed 174 (58.6%) skeletons, while 123 (41.4%) were from general cemeteries or burial sites (Table 2).

## Cranial changes

The number of skeletons that provided data for specific cranial bones ranged from n = 45 for posterior alveolar margins of the maxilla to n = 279 for maxilla (of N = 297 skeletons) (Table A in S1 Table). Pathological changes were most commonly observed in maxilla (247/255, 96.9%), followed by the alveolar process of maxilla (220/233, 94.4%), anterior nasal spine (185/198, 93.4%), palatine process of maxilla oral surface (161/176, 91.4%), palatine process of maxilla nasal surface (168/185, 90.8%), and nasal aperture (141/165, 85.5%); the zygomatic bone had the lowest proportion of pathological

**Table 2. Frequency of skeletons with leprosy (N = 297, from 67 studies) showing sex, age at death and type of burial site.**

|  |  | Frequency | % |
|---|---|---|---|
| **Sex** | Male | 170 | 57.2 |
|  | Female | 81 | 27.3 |
|  | Undefined | 46 | 15.5 |
| **Age*** | Infant | 1 | 0.3 |
|  | Child (<10 years) | 4 | 1.3 |
|  | Adolescent (10–18 years) | 23 | 7.7 |
|  | Adult (≥19 years) | 255 | 85.9 |
|  | Undefined | 14 | 4.7 |
| **Origin** | Leprosarium | 174 | 58.6 |
|  | General Cemetery/Burial | 123 | 41.4 |

* Based on mid-point of estimated age range if no burial record.

alterations at 8.3% (3/36) (Figs 2 and 3). Differences in cranial changes by age were evident only for palatine process of maxilla oral surface, with a higher proportion with irregular perforation of the palate in the ≥ 50 years age group (80%, 8/10) compared with 37.8% (17/45) and 32.3% (20/62) in the 0–24 and 25–49 years age groups, respectively (p = 0.024) (Table B in S1 Table).

### Dental abscesses

Data on dental abscesses in skeletons with leprosy shows that abscesses are most commonly found in the molar region of the maxilla, followed by the premolar region of the maxilla and the molar region of the mandibula (Table C in S1 Table). Across the three age groups, dental abscesses were observed in 10.3% (6/58) of those aged 0–24 years, 18.0% (16/89) of those aged 25–49 years, 21% (4/19) of those aged ≥50 years (Table B in S1 Table), although these differences were not supported by statistical evidence (p = 0.309).

### Nasal structures

The anterior nasal spine was evaluated in 198 skeletons, and resorption and reduction occurred in 107 (54.0%) and 78 (39.4%), respectively (Table A in S1 Table). Nasal aperture was evaluated in 165 skeletons, appearing normal in 24 (14.5%), with progressive smooth resorption with recession of the normal sharp basal lateral margins the most frequent alteration, occurring in 83/165 skeletons (50.3%), followed by progressive smooth resorption with recession of the nasal sharp basal in 32/165 skeletons (19.4%). Presence of pitting only and pitting and progressive resorption with recession of the normal sharp basal and lateral margins were observed in 11/165 (6.7%) and 15/165 skeletons (9.1%), respectively. Pathological alterations at the middle and inferior nasal conchae were found in 39.6% (19/48) and 67% (61/91) of skeletons. The nasal bone was evaluated in 73 skeletons, with pathological alterations observed in 51 (69.9%).

### Rhinomaxillary syndrome (RMS)

Of 279 individuals with sufficient cranial data to assess bone changes defining RMS, 153 (54.8%) were identified as possible RMS, 85 (30.5%) probable RMS, and 41 (14.7%) without RMS. Excluding alterations in the nasal conchae and posterior margins of the maxilla (due to difficulty in identification), 60 out of 85 individuals (70.6%) with probable RMS exhibited a degree of alteration in each of the 5 remaining criteria (I, II, III, IV, and VI).

The age distribution of possible and probable RMS (Table D in S1 Table) indicates that probable RMS was more frequent in the age group ≥50 years (57.9%, 11/19) compared to the 25–49 years (29.7%, 27/91) and ≤24 years groups

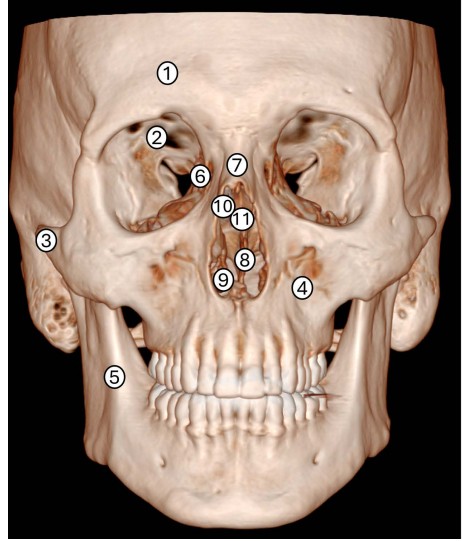

1. Frontal bone (36.7%)
2. Orbital roof (65.2%)
3. Zygomatic bone (8.3%)
4. Maxilla (96.9%)
5. Mandible (73.6%)
6. Lacrimal bone (43.5%)
7. Nasal bone (78.5%)
8. Vomer (80.3%)
9. Inferior nasal turbinate (79.2%)
10. Middle nasal turbinate (54.3%)
11. Perpendicular plate of the ethmoid (78.5%)

**Fig 2. Percentage of pathological alterations of cranial bones in skeletons with leprosy (see Table A in S1 Table for numerator/denominator).**

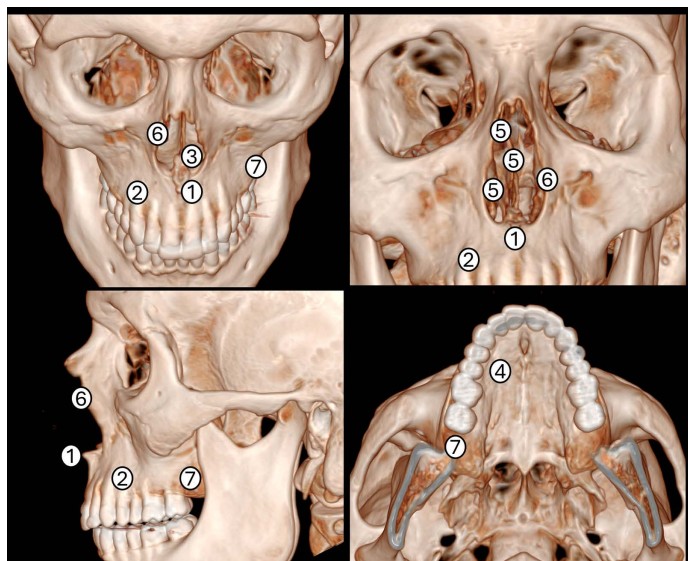

1. Anterior nasal spine (93.4%)
2. Alveolar process of the maxilla (94.4%)
3. Palatine process of the maxilla (nasal surface) (90.8%)
4. Palatine process of the maxilla (oral surface) (91.4%)
5. Nasal turbinates and bony nasal septum (73.2%)
6. Piriform aperture (85.5%)
7. Posterior alveolar margins of the maxilla (61.9%)

**Fig 3. Percentage of pathological alterations of rhinomaxillary bones in skeletons with leprosy (see Table A in S1 Table for numerator/denominator).**

(29.6%, 178/601). However, these apparent differences were not supported by statistical evidence (p = 0.153). There was no difference in RMS by gender, with females constituting 69.6% (32/46) of the non-RMS group and 67.9% (1,320/1,945) of the RMS group (p = 0.739). Similarly, females represented 67.7% (90/133) of skeletons with possible RMS and 65.4% (400/612) of those with probable RMS (p = 0.773).

Possible and probable RMS were equally likely to be observed in remains from general cemeteries and other burial sites (possible 42.1% (48/114), probable 43.0% (49/114)) whereas, in skulls from *leprosaria*, possible RMS was more

frequently observed (63.0% (104/165)) than probable RMS (22.4% (37/165)). The proportions of skulls without RMS were the same in *leprosaria* (14.5% (24/165)) as in non-*leprosaria* sites (14.9% (17/114)).

The oldest skeleton diagnosed with leprosy (individual 257 S20) dated back to 3780–3650 BCE from the *Abony-TurjaÂnyos dűlő* site in Hungary [70]. Despite being negative for *M. leprae* and missing parts of the skull and with fragmented post-cranial bones, this individual met the criteria for probable RMS, showing resorption of the anterior nasal spine, alveolar processes of the maxilla, and piriform aperture, alongside partial post-cranial bone loss. Skeletons '263 S29' and '263 S39' from the same site and period met the criteria for possible RMS due to alterations in the piriform aperture.

### Post-cranial bone changes (PCBC)

PCBC are summarised in Fig 4 (from data in Table E in S1 Table). In leg bones, 14.3% (24/168) of tibias, 14.6% (24/164) of fibulas, and 54.3% (25/46) of femurs showed pathological alterations. In foot bones, 76.6% (85/111) of tarsals, 81.5% (106/130) of metatarsals, and 85.6% (101/118) of foot phalanges exhibited pathological alterations. In arm bones, 75.8% (25/33) of humeri, 65.8% (25/38) of radii, and 61% (25/41) of ulnas exhibited pathological alterations. In hand bones, 27.9% (17/61) of carpals, 45.8% (38/83) of metacarpals, and 75.8% (69/91) of hand phalanges exhibited pathological alterations.

### Correlations of PCBC with age

There were no associations of specific PCBC namely, long bone diaphysis, diaphysis destructive remodelling, and acrosteolysis with age (Tables F-H in S1 Table). Severe bone infections (septic bone changes) like osteomyelitis tended to be slightly more common in older individuals (≥ 50 years old) with leprosy, while younger age groups (0–24 and 25–49 years

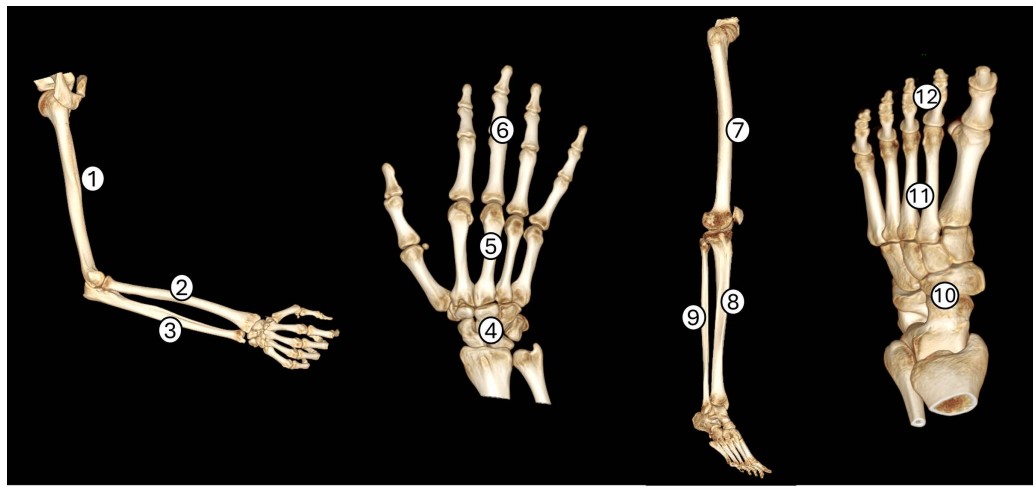

1. Humerus (75.8%)
2. Radius (65.8%)
3. Ulna (61.0%)
4. Carpal bones (27.9%)
5. Metacarpal bones (45.8%)
6. Phalanges of the hand (75.8%)
7. Femur (54.3%)
8. Tibia (14.3%)
9. Fibula (14.6%)
10. Tarsal bones (76.6%)
11. Metatarsal bones (81.5%)
12. Phalanges of the foot (85.6%)

**Fig 4. Percentage of pathological alterations of post-cranial bones in skeletons with leprosy (see Table E in S1 Table for numerator/denominator).**

old) showed a lower incidence of such infections, with periostitis being more frequent in the middle age group (25–49 years old), although numbers are small and apparent differences unsupported by statistical evidence (p = 0.140) (Table I in S1 Table); increasing prevalence of inflammation with age is also likely regardless of leprosy.

**Correlations of RMS with PCBC**

Of the 85 individuals classified as probable RMS cases, 70 had information on PCBC (Fig 5). Of these, 52 individuals (74.3%) exhibited alterations in the PCBC studied in this review, with 7.7% (4/52) showing changes only in the upper limbs, 36.5% (19/52) only in the lower limbs, and 55.8% (29/52) in both upper and lower limbs. Of the 59 individuals with cranial bone alterations corresponding to RMS criteria I, II, III, IV, and VI, 49 had information on PCBC, of whom 42 (85.7%) had at least one PCBC. Associations of long bone diaphysis, diaphysis destructive remodelling, and acroosteolysis with possible or probable RMS are summarised in Tables J-L in S1 Table, respectively. These tables show higher proportions of PCBC in skeletons without RMS, for example, diaphyseal destructive remodelling of tarsals (p = 0.011) and hand phalanx (p = 0.001) (Table K in S1 Table) and acroosteolysis in these bones (p = 0.003, p = 0.010, respectively) and in metatarsals (p = 0.050) and metacarpals (p = 0.020) (Table L in S1 Table), but this statistical evidence needs to be interpreted in the context of small numbers of bones available for analysis and multiple statistical tests.

Overall, acroosteolysis was observed in feet localised in 70.6% (77/109) of tarsals, 78.9% (101/128) of metatarsals, and 83.6% (97/116) of foot phalanges. In hands, acroosteolysis was found in 25.0% (15/60) of carpals, 44.4% (36/81) of metacarpals, and 75.8% (69/91) of hand phalanges (Table L in S1 Table).

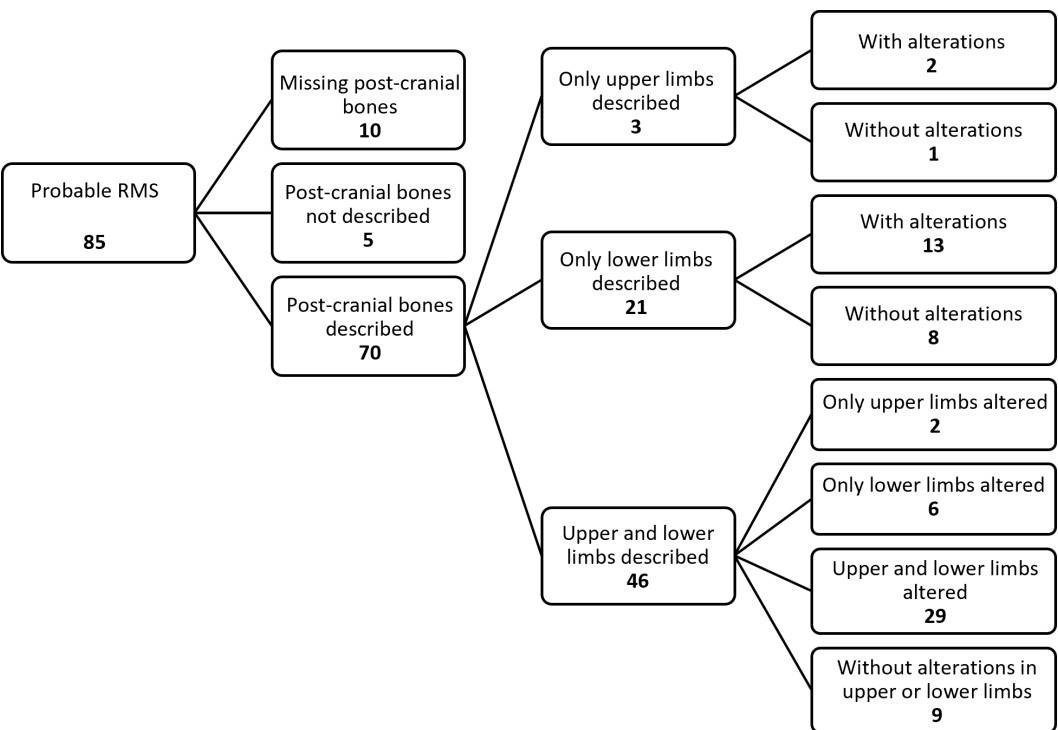

**Fig 5. Post-cranial bone changes in skeletons with probable rhinomaxillary syndrome (RMS).**

## Skeletons from the region of modern-day Europe

Skeletons from Europe were dated from 3715 BCE to 1839 CE (Fig 6), with a peak around the 15th century corresponding mainly to studies of skeletons from Denmark (St. Jorgen's Hospital, Naestved, n=67) [5–7,47,48] and the UK (Hospital of St. James and St. Mary Magdalene, Chichester, n=86) [55], where skeletons were dated from 1250-1550 CE and 1300–1700 CE, respectively. Of the 264 skeletons from Europe, 54 (20.5%) presented only PCBC, 68 (25.8%) probable RMS, and 142 (53.8%) possible RMS. Distribution over time suggests relatively constant proportions of possible or probable RMS during the peak periods covered by archaeological investigations (1301 CE to 1500 CE), with approximately 1 in 5 skeletons without RMS, a similar proportion with probable RMS, and half to two-thirds showing possible RMS. In the earlier period (-3800 BCE to 1300 CE), the proportions with probable RMS or without RMS were higher, while in the later period (1501 CE to 1900 CE), the proportion with possible RMS was higher (three-quarters of skeletons) and the proportion without RMS was lower (fewer than 1 in 10) (Table 3).

## Genomic studies

The 19 genomic studies [12,13,50,75,107–121], of which 7 were from sites outside Europe (2 Egypt [110,118], 1 Japan [100], 1 Russia [111], 1 Suriname [102], 1 Turkey [115] and 1 Uzbekistan [116]) described *M. leprae* genotypes in 73 skeletons (Fig 7), most of which had previously been described in an earlier non-genomic study. Of the 73 skeletons analysed, single- nucleotide polymorphism (SNP) type 1 was identified in one, SNP subtype 2F in 11 (15.1%), SNP type 3 in 59 (80.8%), and SNP types 3Q and 4 each in one (S1 Appendix). Of the 59 SNP type 3, 13 were subtype 3I, 18 were 3I-1, 11 were 3K, two were 3L and two were 3M. Six of the genomic studies attempted to identify *M. lepromatosis*, but none yielded positive results [50,75,102,111,117,121].

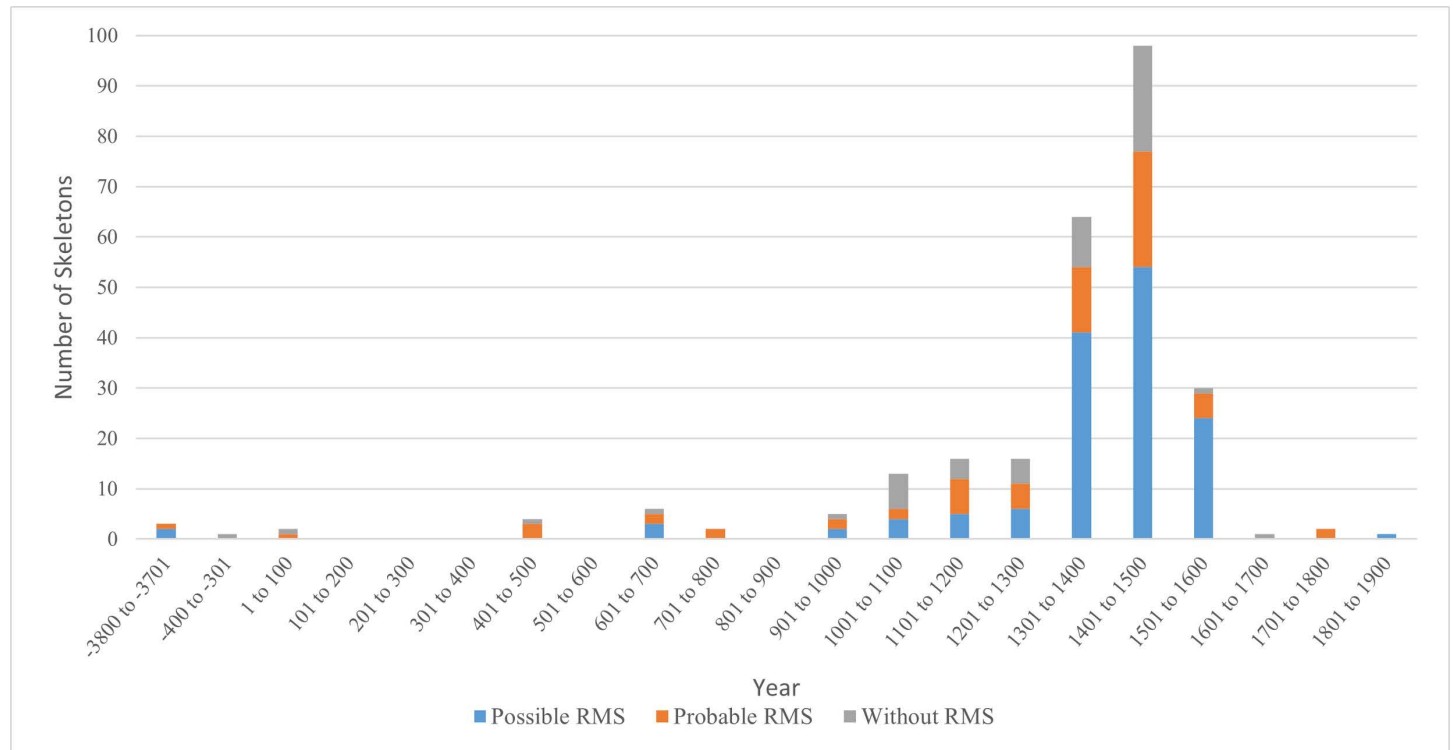

**Fig 6. Rhinomaxillary syndrome (RMS) by century of dating of skeletons from archaeological sites in Europe\*.** \* Before Common Era (BCE) centuries are shown on horizontal axis only if remains were dated during that century.

**Table 3. Probable and possible rhinomaxillary syndrome (RMS) from archaeological sites in Europe by century of dating (N = 264 skeletons from 53 studies).**

| Century | Possible RMS | Probable RMS | Without RMS |
|---|---|---|---|
| -3800–1300 | 22 (32.4%) | 25 (36.8%) | 21 (30.9%) |
| 1301–1400 | 41 (64.1%) | 13 (20.3%) | 10 (15.6%) |
| 1401–1500 | 54 (55.1%) | 23 (23.5%) | 21 (21.4%) |
| 1501 to 1900† | 25 (73.5%) | 7 (20.6%) | 2 (5.9%) |
| Overall | 142 (53.8%) | 68 (25.8%) | 54 (20.5%) |

† Rows combined due to small numbers: 1501–1600, possible n = 24 (80.0%), probable n = 5 (16.7%), without RMS n = 1; 1601–1900, possible n = 1, probable n = 2, without RMS n = 1.

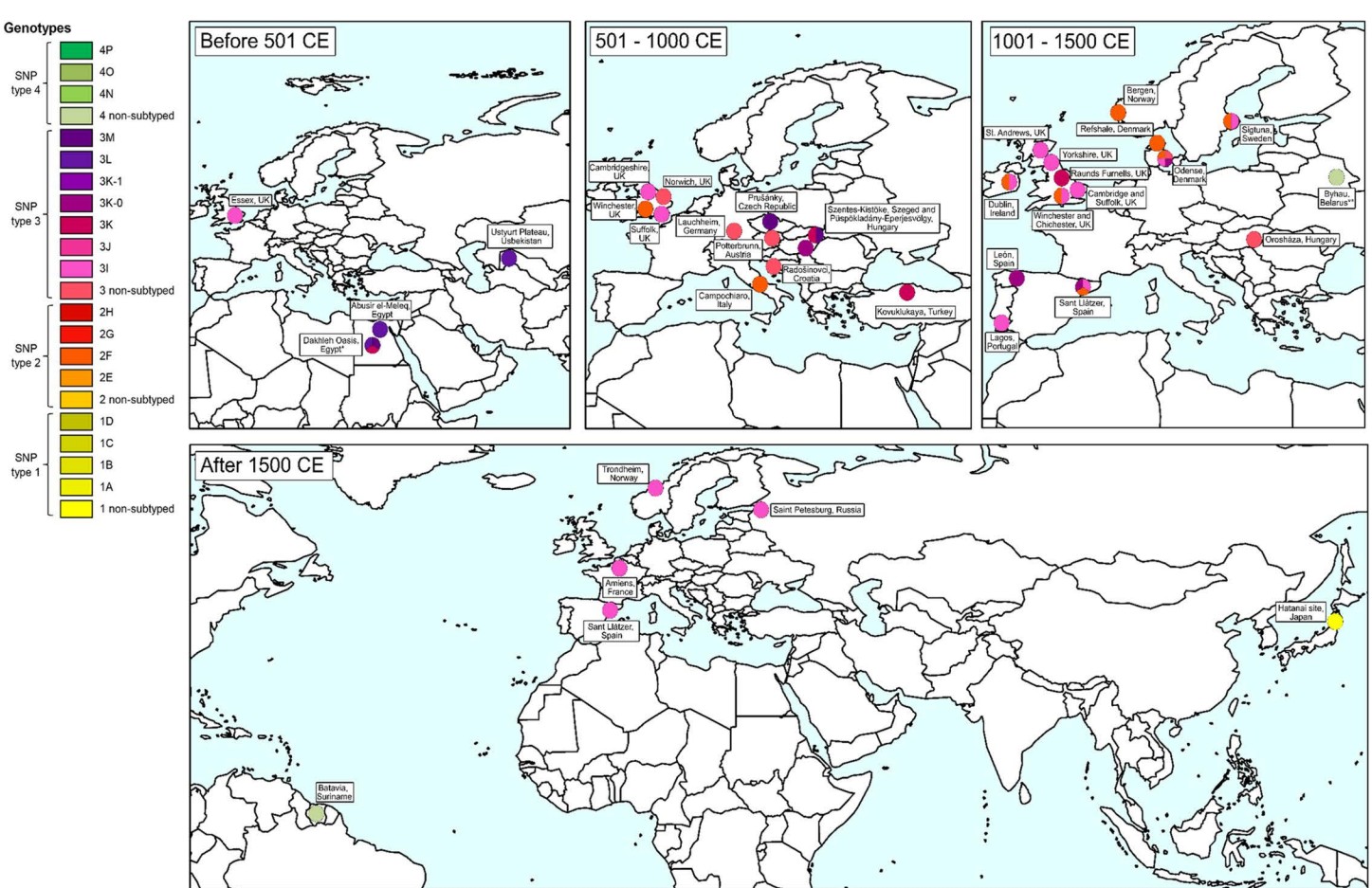

**Fig 7. Geographical distribution of *Mycobacterium leprae* genotypes from archaeological sites.** * Genotype from Ikeleton K2-B116 did not show an exact match. ** Probable new Subtype (3Q). Source: this figure was created in ArcGIS and its shapefile was obtained in Natural Earth via the following link: https://www.naturalearthdata.com/downloads/110m-cultural-vectors/110m-admin-0-countries.

In 2018 [13], a skeleton described in 2005 [78] (individual T.74, 350–300 BCE) from Bologna, Italy was the oldest to test positive for *M. leprae*. In 2015 [12], SNP type 3L was identified in a skeleton (individual Kurgan 5b, 80–240 CE) from Devkesken, Uzbekistan, first described in 2005 [105], becoming the oldest remains to be genotyped. The oldest

European skeleton to be genotyped (individual CG96, 415–545 CE, SNP type 3I) was from Great Chesterford, Essex, United Kingdom [54].

## Discussion

This systematic review provides an in-depth examination of the geographic and chronological distribution of paleopathological cases of leprosy, offering a perspective on the disease's historical impact across various continents, albeit mainly in the region of modern-day Europe where archaeological sites were concentrated. In addition to mapping archaeological sites where remains with leprosy were studied, the review provides a thorough analysis of bone alterations, both cranial and post-cranial. Paleopathological studies have long depended on these physical alterations to identify cases, and our review highlights the importance of *M. leprae* ancient DNA and genotyping analyses in providing confirmatory evidence for individual skeletons and in exploring hypotheses for the decline of leprosy in Europe. Beyond attempting to elucidate the historical trajectory of leprosy, paleopathological research has relevance to present-day societies where the disease remains endemic, including contextualising societal attitudes to a disease which still carries a burden of stigma and where outmoded public health responses such as segregation of affected persons live in recent memory [122].

Direct examination of bone changes in paleopathology specimens can improve understanding of clinical manifestations of *M. leprae* infection in living persons affected by Hansen's disease [41]. For example, we previously used computed tomography (CT) scans to assess the prevalence of RMS among elderly former patients of a 'colony' hospital in Brazil [40]. We also used CT imaging combined with otorhinolaryngological examinations to assess maxillofacial and nasal manifestations of leprosy in current patients [123]. Findings from these studies suggest that clinical protocols for Hansen's disease could be extended to include otorhinolaryngological evaluation, supported by imaging where necessary, to improve the assessment and overall care of persons affected by Hansen's disease. This systematic review synthesizes evidence from palaeopathological studies, characterising cranial and post-cranial bone alterations associated with leprosy, serving as a reference point for clinical research and practice today.

### Geographical and temporal distribution of leprosy genotypes

Phylogenetic analysis has revealed a high degree of genomic conservation within *M. leprae*, characterized by four primary SNP types divided into at least 16 subtypes [118]. This allows analyses of ancient DNA to generate hypotheses regarding the geographic and temporal spread of the disease over historical time period, Prior to 501 CE, leprosy was primarily found in the Middle East and parts of Europe, with SNP genotypes 3I and 3L recorded in regions such as the Dakhleh Oasis [118] and Abusir el-Meleq [110] in Egypt. Between 501 and 1000 CE, the disease expanded into Central and Northern Europe, evidenced by genotypes such as 3K, 3M and 2F in areas like Prušánky, Czechia [13], and Lauchheim, Germany [66]. From 1001 to 1500 CE, there was an increased presence of leprosy in Western Europe, with a predominance of branch 3 strains in Western and late Medieval Europe, with branch 2 the next most common strain, but against a background of high genetic diversity of *M. leprae* across the region and even within *leprosaria* [111,124]. Accordingly, our review found that the majority of identified *M. leprae* genotypes were SNP type 3, accounting for 81% of cases, followed by SNP type 2 (15%). However, the small numbers of cases that have been genotyped across a large geographic region that witnessed continual movement and mixing of people over a long time period only serve to generate rather than test hypotheses regarding the phylogeography of *M. leprae*.

### Rhinomaxillary syndrome in leprosy

Although RMS was initially identified based on Andersen and Manchester's seven criteria, this review scrutinized cases by examining the presence or absence of a subset of cranial changes that characterize possible or probable RMS. The definition of probable and possible LL based on a subset of the RMS criteria was an adaption from a study carried out in

the Paris Catacombs, where skulls were fixed in position providing access only from the front [64]. In the present review, in which we have access only to secondary information (data extracted from published studies), we adopted the same subset as synonymous with probable and possible RMS. This is because we consider criteria I (resorption of the anterior nasal spine), II (resorption of the alveolar processes of maxilla), and VI (enlarged pyriform aperture) as the key cranial bone alterations in leprosy where information is limited, and RMS is likely to occur only in severe (LL) forms of leprosy, as we see in present day clinical studies [40,41]. The pathognomonic nature of RMS as described by Andersen & Manchester [38] is not universally accepted, because other diseases of relevance to paleopathology, including venereal syphilis, tuberculosis, mucocutaneous leishmaniasis, and malignant tumors can lead to maxillofacial and cranial bone lesions [125]. Our review found that approximately 80% of skeletons from archaeological sites across various centuries exhibit some degree of RMS. Notably, RMS appeared more frequently in remains from the 16th century onwards (94%) compared to earlier periods, as has been observed in recent decades as countries move towards elimination [126,127].

Three-quarters of the skeletons diagnosed with RMS also presented PCBC. The presence of PCBC suggests a very advanced stage of the disease with grade 2 disability (G2D) as suggested by Moller-Christensen [5]. The most significant PCBC affect the tubular bones of the hands and feet. These lesions manifest as acroosteolysis and destructive remodelling of diaphysis [7,9,128,129]. Although less specific regarding aetiology, osteomyelitis, interphalangeal fusions, tarsal disintegration, secondary fractures and proliferative lesions in the tibia and fibula are also observed [130,131]. Associations of PCBC with age and sex were not evident, possibly due to the limited sample sizes. Conversely, among 70 individuals with RMS and available data on the bones of the hands and feet, 18 exhibited RMS without corresponding alterations in these bones. Although limited by modest numbers of skeletons, we found some evidence for higher proportions of PCBC in skeletons without RMS, for example, diaphyseal destructive remodelling of tarsal and hand phalanx and acroosteolysis in these bones and in metatarsals and metacarpals. Asymmetric PCBC without RMS may indicate a tuberculoid or borderline tuberculoid case, while symmetric PCBC without RMS may suggest a multibacillary borderline form, such as borderline-borderline or lepromatous-borderline, with the caveat that small numbers of bones were available for analysis and we performed multiple statistical tests.

Our finding that 54% of femurs were reported as affected appears atypical when compared to the lower proportions of affected tibias (14%) and fibulas (15%). Typically, skeletal lesions involve the small bones of the hands and feet, the nasal region, and the tibiae due to their subcutaneous location and proximity to cooler areas of the body, where *M leprae* tends to localize. Given the relatively small number of femurs examined (n = 46), the high percentage (54%) may reflect sampling bias, preservation bias, or reporting bias in the available osteoarchaeological studies. For instance, femurs may be better preserved or more frequently recovered than smaller bones in certain burial contexts or studies may have focused on femoral lesions when they occurred given their relative rarity.

According to Møller-Christensen [5], cranial bone changes associated with leprosy were strongly correlated with nasal alterations, such as an increased nasal aperture in 100% of cases, and atrophy of the anterior nasal spine in 76%. Additionally, *usura orbitae* was observed in 63% of individuals who had inflammation of the nasal cavity. The findings in this review emphasise changes in the nasal area as indicators of leprosy in skeletal remains, with 70–80% of nasal bones showing pathological alterations.

### Cases of leprosy in childhood

Among the skeletons assessed, 12 were of adolescent age and six were under 10 years old [5,7,58,81], including two skeletons age under 5 years old at the time of death [81]. The youngest was an infant age 4–5 months who, although harbouring *M. leprae* aDNA in the occipital bone, lacked skeletal manifestations of leprosy. This indicated an absence of clinical signs typically associated with the disease. It is important to note that the extraction and analysis of leprosy aDNA from skeletons does not necessarily imply that these individuals experienced overt clinical symptoms of the disease; they may have had subclinical leprosy without apparent signs or symptoms of infection [106]. The skeleton identified as a child aged 4–5 years old exhibited profound cranial alterations, including complete resorption of the anterior nasal spine,

intense erosive activity in the alveolar processes of the maxilla, and bilateral symmetrical resorption and remodelling of the piriform aperture. These features align with possible RMS and indicate an advanced lepromatous form of the disease. Advanced bone changes due to LL, such as RMS, are exceptionally rare in young individuals. Among the six cases studied, a plausible explanation could involve genetic susceptibility to *M. leprae* infection coupled with significant exposure to high bacillary loads, potentially exacerbated by conditions such as severe malnutrition or underlying bone malformations [132].

*Leprogenic odontodysplasia* is a manifestation of multibacillary leprosy that affects the normal development of tooth roots, particularly of the maxillary incisors, in conjunction with RMS, described first by Danielsen in the skeletons of four children aged 8–11 years from the medieval leprosy cemetery at Næstved, Denmark [133]. There are no records of clinical cases. We did not extract data on *leprogenic odontodysplasia*, but three of the studies included in our review described other cases, namely: a child aged 9–11 years from the St Mary Magdalen leprosy hospital in England [61], a juvenile aged 13–19 years from St. Jørgen's *leprosarium* cemetery at Odense, Denmark [48], and a child aged 11–12 years from Sigtuna, Sweden [89].

## Decline of leprosy in Europe

The limited and selective nature of data from archaeological sites cannot substantiate the various theories regarding the disappearance of leprosy in Europe. Studies that have reported *M. leprae* and *M. tuberculosis* co-infection [17,18] lend some support to epidemiological models that suggest that, as the prevalence of tuberculosis increased, the prevalence of leprosy may have declined due to temporal changes in immune status influenced by bacterial exposure [128], but this hypothesis is complicated by gaps in our contemporary understanding of cross-immunity [134], including with other non-tuberculous mycobacterial species in the environment [135], and by the impact of other historical factors such as urbanization [136] and population movements [19]. Testing such hypotheses is constrained by the small numbers of geographically and temporally dispersed remains for paleopathological research, as characterised in our review.

In South Korea, where leprosy has been eliminated, the process revealed a shift towards an increased proportion of multibacillary cases and older age groups [126]. Data from Norway's epidemiological records show an increase in the proportion of affected persons over 50 years old preceding leprosy elimination [127]. A decline in leprosy driven by a shift from the paucibacillary forms (tuberculoid and borderline tuberculoid forms) towards the multibacillary end of the leprosy spectrum (borderline-borderline, lepromatous-borderline and LL) would manifest in skeletal evidence in a concomitant increase of RMS cases. Our review suggested more frequent RMS in the relatively small number of remains from the 16th century onwards, but this was unsupported by statistical evidence and may reflect the state of preservation of remains and increased longevity (hence, more time for disease to progress).

*Leprosaria* in Europe have been crucial sites for the advancement of paleopathological research on leprosy. In our data there was a predominance of burials in non-leprosarium cemeteries across the earlier time periods, with a notable increase in leprosarium burials beginning around the 11th century CE, peaking during the 14th to 16th centuries, and reflecting the increasing number of specialized leprosy 'hospitals' during this period. Roberts suggests that the rise of *M. leprae* infections in Europe after the 10th century, followed by their decline after the mid-16th century, was neither gradual nor uniformly distributed, indicating regional differences in the impact and influence of underlying factors [11]. As we saw in our review, much of the palaeopathological evidence on European leprosy has been generated from three medieval *leprosaria* – Chichester [55] and Winchester [58] in England, and Naestved [5] in Denmark – which limits broader regional generalisations.

Limitations of data aggregated from paleopathological studies include varying states of preservation of human remains and the absence of radiocarbon dating for many skeletons. The lack of a standardized investigative protocol for identifying leprosy-related bone changes means that the secondary data summarised in our review may be susceptible to intra- and inter-observer error made by the original researchers. We presented post-cranial bone changes in aggregate, but significant variability likely exists in the recovery and documentation of hand and foot bones, influenced by the specific

objectives of archaeological excavations and the availability of osteological expertise at different sites. Although the location and type of archaeological site might be expected to influence the types of bone alterations in remains, we found equal proportions (15%) of skulls without RMS from leprosaria and from general cemeteries or other burial sites. This observation challenges the assumption that *leprosaria* served as focal destinations and final resting places for people in advanced disease stages, who would presumably exhibit distinctly identifiable facial deformities [41,40] or that individuals within *leprosaria* necessarily experienced prolonged survival, allowing more time for such deformities to develop. However, we also recognise that the entire paleopathological population in our review is pre-selected for the more severe forms of leprosy that result in visible bone changes, regardless of site of exhumation, which explains the overall high frequency of RMS in our study.

Despite these challenges, the data presented in our review are consistent with leprosy being present in the European region for over five millennia, with prevalence increasing after the 10th century and peaking around the 14th-15th centuries. After this period, the number of skeletons presenting signs of the disease abruptly decreased. Notably, no skeleton was retrospectively diagnosed and dated after 1850 in Europe.

## Leprosy outside Europe

Leprosy is endemic in several countries of South Asia and South America, with 80% of the 200,000 new cases each year diagnosed in India, Indonesia and Brazil [3,4]. There are no descriptions of leprosy in the pre-colonial American continent, consistent with importation of the disease from Europe and Africa during that period [137]. However, one of the few individuals retrospectively diagnosed with leprosy in the 19th-20th centuries was an adolescent, of unknown sex, from Suriname, with cranial and post-cranial alterations compatible with advanced leprosy [102]. Even though the skeleton dates from 1850-1900 CE, it represents the oldest archaeological identification of *M. leprae* in the Americas, with aDNA analysis positive for SNP type 4 *M. leprae* and identification of mitochondrial haplogroup L3 revealing a genetic ancestry from African and/or Middle Eastern populations [102,138].

Studying leprosy as an ancient disease is important for several reasons. It enhances our understanding of the evolution of leprosy and provides insights into how the disease has changed over time, including variations in its manifestations and impact on different populations. Such research also contributes to mapping the historical and potential future global distribution of leprosy, particularly in relation to climate change, human migration, and disease control efforts. One aspect of paleopathological research of relevance to the present day is the role of animal and environmental reservoirs of *M. leprae* (and *M. lepromatosis*) and the potential for zoonotic transmission to introduce and sustain infection in human populations [139]. In present day South America, particularly in Brazil where Hansen's disease remains endemic, *M. leprae* is prevalent in armadillos, and human contact through hunting and consumption is common, there is a compelling argument for adopting a One Health approach to Hansen's disease [140], whilst further evidence is needed to assess the zoonotic potential of *M. lepromatosis* in the region [141]. Phylogenetic evidence from medieval Europe reinforces these links, with a study from Winchester in England showing a close relationship between ancient *M. leprae* squirrel and human strains and suggesting that *M. leprae* circulated in non-human hosts in the Middle Ages [142]. Moreover, strains from the Late Medieval period are genetically similar to those now circulating in humans and armadillos in the Americas [111]. These findings underscore how palaeopathological research clarifies historical transmission patterns that remain relevant today, supporting integrated, cross-disciplinary strategies to control leprosy in endemic settings.

The study of leprosy in historical populations presents unique methodological challenges that necessitate continuous refinement of research techniques. Current limitations include the degradation of DNA in ancient samples and the ethical and legal constraints surrounding destructive sampling, and the difficulty in distinguishing leprosy from other diseases based solely on skeletal evidence. Future investigations should focus on improving methods for extracting and analyzing ancient DNA, potentially incorporating next-generation sequencing technologies to provide deeper insights into the genetic makeup of *M leprae*. Newer imaging modalities such as micro-CT may allow for deeper characterisation of bone

alterations and provide a way of confirming leprosy diagnosis in fragmented remains [143]. These techniques may also support age-matched comparisons between individuals from *leprosaria* and those from general burial sites, enabling researchers to explore the effects of institutional care, treatment access, and disease progression on skeletal pathology. Additionally, interdisciplinary approaches combining archaeology, genetics, radiology, and historical documentation can enhance our understanding of how environmental factors and human migration have influenced the spread and evolution of leprosy. By addressing these challenges, researchers can better reconstruct the historical epidemiology of leprosy and contribute to more effective public health strategies in regions where the disease is still endemic.

In conclusion, the paleopathological evidence synthesised in this systematic review offers insights into the historical distribution and severity of bone changes associated with leprosy, with a focus on cranial changes pathognomonic of the 'lepromatous' form of the disease. The integration of genetics and, potentially, biomolecular approaches such as those based on mycolipids, proteins and peptides into paleopathology, will be essential for validating the diagnoses in archaeological specimens and for elucidating the complex interactions between *M. leprae* and human populations throughout history..

## Methods

### Databases and searches

Searches were conducted in PubMed, Google Scholar, EMBASE, and Scopus electronic databases using terms including "leprosy", "Hansen's disease", "paleopathology", "archaeology", "burial", "leprosarium", "skeletal", "skeletal lesions", "remains", and "medieval". We also searched references in the bibliographies of retrieved articles. Searches were performed in July 2022 and updated in December 2023.

### Inclusion and exclusion

We included articles in Portuguese, English, Spanish, and French that characterized bones with lesions caused by leprosy, with or without the presence of a causative agent. We excluded articles in which the presence of another disease may have caused the lesions. We excluded review articles but searched the bibliographies of these articles to identify further references for potential inclusion.

### Data extraction

Data were extracted independently and in parallel by two reviewers. Extracted data were tabulated in Excel in two parts: a) study identifiers, excavation site(s), and other archaeological information; b) skull and bone alterations in individual skeletons, including estimated age at death, year of death (from burial record) else range, sex, presence of bone lesions characteristic of leprosy, *M. leprae* genotype findings, and other observations related to each individual. A separate round of data extraction was performed for studies that reported new genetic analyses of previously described individuals.

### Quality assessment

Quality assessment was performed independently and in parallel by two additional reviewers using separate instruments for original archaeological studies and for studies that reported new genomic findings from these studies. Quality was assessed using a scoring system whereby a score of 1, 0.5, or 0 was assigned to each of 8 criteria (6 for genomic studies) if it was fully, partially, or not met, respectively (S1 Appendix). Original references were graded as A (>6 points), B (>3 and ≤6 points), and C (≤3 points), and genomic references as A (>4 points), B (>2 and ≤4 points), and C (≤2 points), with grade C references excluded from further analyses.

### Identification of rhinomaxillary syndrome (RMS)

The presence of RMS was identified from extracted data according to Andersen and Manchester's seven criteria (Box 1 and S1 Fig). We used a previous adaptation of these criteria [64] to classify skulls as showing 'probable' RMS if they had

an enlarged nasal (piriform) aperture, resorption of the anterior nasal spine, and resorption of the alveolar processes of maxilla, or 'possible' RMS if they met one or two of these three criteria (Box 1). Post-cranial bone changes (PCBC) were identified according to the criteria presented by Roberts [144] (Box 2).

**Data analysis**

Data were described as frequency (%) and plotted by century and age group. When studies reported a range of dates for remains, we used the midpoint for our analysis of trends over centuries in Europe. Our definition of Europe excluded the Eurasian countries, e.g., Turkey, Georgia, Armenia. Differences in frequencies of bone alterations by the presence or absence of RMS and by age group were tested in Stata (StataCorp. 2023. Stata Statistical Software: Release 18. College Station, TX: StataCorp LLC.) using Fisher's exact test (α = 0.05).

---

### Box 1. Rhinomaxillary syndrome in leprosy paleopathology (see also S1 Fig).

| Criterion | Bone alterations |
|---|---|
| I | Anterior nasal spine: resorption and ultimate loss with exposure of medullary bone followed, possibly, by cortical remodelling. |
| II | Alveolar processes of maxilla: bilateral and symmetrical resorption and recession commencing centrally at the prosthion and extending to the alveolae of the central and lateral incisors and canines, with loss of these teeth. |
| III | Nasal surface of the palatine process of the maxilla: inflammatory change leading to localised bone destruction and ultimate perforation of the palate, usually in the median or paramedian position. |
| IV | Oral surface of the palatine process of the maxilla: inflammatory change leading to localised bone destruction and ultimate perforation. |
| V | Conchae (turbinate bones) and nasal septum: inflammatory pitting with or without slight irregular periosteal new bone formation or destruction and ultimate loss of the bony nasal septum, and loss of one or more conchae. |
| VI | Piriform aperture: progressive smooth resorption with recession of the normally sharp basal and lateral margins of the piriform aperture, inferiorly. |
| VII | Posterior alveolar margins of the maxilla: resorption in the region of the molar teeth, commencing at the third molars. |
| I or II or VI | **Possible RMS** |
| I and II and VI | **Probable RMS** |

---

### Box 2. Post-cranial bone changes in leprosy paleopathology.

| Location | Bone alterations |
|---|---|
| Hands | Carpal, metacarpal, and phalanges: bone extremities resorption (acroosteolysis) and diaphyseal destructive remodelling lesions, such as cortical bone resorption (progressive loss of diameter), concentric diaphyseal remodelling (hourglass appearance), and remodelling, shortening or destruction of distal bones ("knife-edge", "shark-tooth", "licked candy stick", "cup-and-peg" appearances). |
| Upper Limbs (Long Bones) | Humerus, radius, and ulna: presence of septic bone changes (periostitis, osteitis, osteomyelitis, and septic arthritis) and superficial inflammation leading to periosteal new bone formation. |
| Feet | Tarsal, metatarsal, and phalanges: bone extremities resorption and diaphyseal destructive remodelling lesions, such as cortical bone resorption, concentric diaphyseal remodelling, and remodelling, shortening, or destruction of distal bones. |
| Lower Limbs (Long Bones) | Tibia, fibula and femur: presence of septic bone changes (periostitis, osteitis, osteomyelitis, and septic arthritis) and superficial inflammation leading to periosteal new bone formation. |

## Supporting information

**S1 Fig. Bone alteration criteria defining rhinomaxillary syndrome in leprosy paleopathology (see Box 1 in main article).**
(TIF)

**S1 Appendix. Included references.** Quality assessment tool. Quality assessment ratings. Data extraction. Included references (genomic). Quality assessment tool (genomic). Quality assessment ratings (genomic). Data extraction (genomic).
(XLSX)

**S1 Table. Table A.** Bone lesions in skeletons with leprosy (N = 297). **Table B.** Bone lesions in skeletons with leprosy by age group (N = 171). **Table C.** Position and location of dental abscesses in skeletons with leprosy (N = 60). **Table D.** Possible and probable rhinomaxillary syndrome (RMS) in skeletons with leprosy by age group (N = 167). **Table E.** Presence of any pathological alterations in post-cranial bones in skeletons with leprosy (N = 297). **Table F.** Long bone diaphysis in skeletons with leprosy by age group (N = 171). **Table G.** Diaphyseal destructive remodelling in hand and foot bones from skeletons with leprosy by age group (N = 171). **Table H.** Acroosteolysis in hand and foot bones from skeletons with leprosy by age group (N = 171). **Table I.** Septic bone changes in skeletons with leprosy by age group (N = 171). **Table J.** Long bone diaphysis in skeletons with leprosy by presence/absence of possible or probable rhinomaxillary syndrome (RMS) (N = 287). **Table K.** Diaphyseal destructive remodelling in hand and foot bones from skeletons with leprosy by presence/absence of possible or probable rhinomaxillary syndrome (RMS) (N = 287). **Table L.** Acroosteolysis in hand and foot bones from skeletons with leprosy by presence/absence of possible or probable rhinomaxillary syndrome (RMS) (N = 287).
(XLSX)

**S1 Checklist. PRISMA 2020 Checklist.**
(DOCX)

## Author contributions

**Conceptualization:** Hugo Pessotti Aborghetti, Patrícia D. Deps.

**Data curation:** Hugo Pessotti Aborghetti.

**Formal analysis:** Hugo Pessotti Aborghetti, Simon M. Collin, Patrícia D. Deps.

**Investigation:** Hugo Pessotti Aborghetti, Simon M. Collin, Julienne Dadalto dos Santos, Pamela Barbosa dos Santos, Taís Loureiro Zambon, Rafael Maffei Loureiro, Patrícia D. Deps.

**Methodology:** Hugo Pessotti Aborghetti, Simon M. Collin, Rafael Maffei Loureiro, Patrícia D Deps.

**Supervision:** Simon M. Collin, Patrícia D. Deps.

**Validation:** Simon M. Collin, Julienne Dadalto dos Santos, Pamela Barbosa dos Santos, Taís Loureiro Zambon, Rafael Maffei Loureiro, Patrícia D. Deps.

**Writing – original draft:** Hugo Pessotti Aborghetti, Simon M. Collin, Patrícia D. Deps.

**Writing – review & editing:** Hugo Pessotti Aborghetti, Simon M. Collin, Julienne Dadalto dos Santos, Pamela Barbosa dos Santos, Taís Loureiro Zambon, Rafael Maffei Loureiro, Patrícia D. Deps.

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
