## [Decision Letter · Decision Letter 0]

18 Mar 2025

PNTD-D-24-01634

LEPROSY IN SKELETONS FROM ARCHAEOLOGICAL SITES: A SYSTEMATIC REVIEW

Dear Dr. Deps,

Thank you for submitting your manuscript to PLOS Neglected Tropical Diseases. After careful consideration, we feel that it has merit but does not fully meet PLOS Neglected Tropical Diseases's publication criteria as it currently stands. Therefore, we invite you to submit a revised version of the manuscript that addresses the points raised during the review process.

Please submit your revised manuscript within 60 days May 17 2025 11:59PM. If you will need more time than this to complete your revisions, please reply to this message or contact the journal office at plosntds@plos.org. Please include the following items when submitting your revised manuscript:

We look forward to receiving your revised manuscript.

Kind regards,

Katharina Röltgen

Academic Editor

Mathieu Picardeau

Section Editor

Shaden Kamhawi

co-Editor-in-Chief

Paul Brindley

co-Editor-in-Chief

**Additional Editor Comments:**

We have received three very detailed and insightful reviews from experts in the field regarding your paper. They have raised several important concerns and provided valuable suggestions for improving the study design, data and statistical analyses, and critical engagement with the limitations of palaeopathological data. We kindly request that you address both the major and minor comments made by the three Reviewers in a revised version of your article, as these revisions will be necessary before the paper can be considered for publication in PLoS NTDs.

**Journal Requirements:**

At this stage, the following Authors/Authors require contributions: Hugo Pessotti Aborghetti, Simon M Collin, Julienne Dadalto dos Santos, Pamela Barbosa dos Santos, Taís Loureiro Zambon, Rafael Maffei Loureiro, and Patricia Duarte Deps. Please ensure that the full contributions of each author are acknowledged in the "Add/Edit/Remove Authors" section of our submission form.

5) We have noticed that you have uploaded Supporting Information files, but you have not included a list of legends. Please add a full list of legends for your Supporting Information files after the references list.

Potential Copyright Issues:

- Please confirm that you are the photographer of Figures 2, 3, 4, and Figure 1 - PRISMA flow diagram, or provide written permission from the photographer to publish the photos under our CC BY 4.0 license.

- Figure 7. Please provide a direct link to the base layer of the map (i.e., the country or region border shape) and ensure this is also included in the figure legend; and provide a link to the terms of use / license information for the base layer image or shapefile. We cannot publish proprietary or copyrighted maps (e.g. Google Maps, Mapquest) and the terms of use for your map base layer must be compatible with our CC BY 4.0 license.

**Reviewers' Comments:**

Reviewer's Responses to Questions

**Key Review Criteria Required for Acceptance?**

**Methods:**

-Are the objectives of the study clearly articulated with a clear testable hypothesis stated?

-Is the study design appropriate to address the stated objectives?

-Is the population clearly described and appropriate for the hypothesis being tested?

-Is the sample size sufficient to ensure adequate power to address the hypothesis being tested?

-Were correct statistical analysis used to support conclusions?

-Are there concerns about ethical or regulatory requirements being met?

Reviewer #1: The aims and methods and approaches taken are clearly outlined. The MS is a revue article of the palaeopathology and phylogeography of leprosy due to M. leprae through time and as such is ambitious. It is necessarily governed by several pre-existing limitations; the archaeological record available Genomic studies are further restricted by permissions for destruvtive sampling, DNA survival and current aDNA techniques obtaining at their time of study.

Reviewer #2: The objectives of the study are clearly articulated with clearly testable hypothesis.

The systematic review is well-defined. The only critical point is that there is a discrepancy between the 86 in the systematic review and the overall 134 references. 40% of the references are not accounted for. TYhe authors should mention that the other set of references are included on the basis of snow-balling or another approach.

Reviewer #3: There needs to be a clearer objective complete with hypothesis for testing. This will improve the study design, structure of results and study design. The authors also need to consider whether they have enough material to support some of their conclusions, e.g. you can't talk about the prevlance of a strain of a disease when n is low or even 1. The statistical analysis for what they did was fine, but it lacks detail and is very basic.

In addition, there has been limited engagement with the limitations of palaeopathological data which impacts on the soundness of the conclusions.

**Results:**

-Does the analysis presented match the analysis plan?

-Are the results clearly and completely presented?

-Are the figures (Tables, Images) of sufficient quality for clarity?

Reviewer #1: The Results are clearly presented in the main paper with Tables, flow diagrams and Figures. This is supplemented by a great deal of additional and detailed information in the Appendices and supplementary Tables in excel spreadsheet format. There are some inconsistencies and gaps in the latter section which could be corrected, and specific suggestions have been made in the main review for the authors' consideration. See Summary and General Comments below.

Reviewer #2: The results are clearly and completely represented

Reviewer #3: Generally yes, but the aims and objectives are rather loose. The results are sometimes incomplete. There are no statistical tests results presented in the text and it is unclear when they were used. In addition, there is a lack of detail about whether correction factors were used or not.

When presenting skeletal data we are not always presented with observation values, the hand and foot data is particularly problematic and it is hard to know whether it is meaningful or not. There is use of vague terminology (see comments below).

**Conclusions:**

-Are the conclusions supported by the data presented?

-Are the limitations of analysis clearly described?

-Do the authors discuss how these data can be helpful to advance our understanding of the topic under study?

-Is public health relevance addressed?

Reviewer #1: (No Response)

Reviewer #2: The conclusions are concise and well-structured.

Reviewer #3: The conclusions are weak really since the data is a little problematic (see comments below).

The limitations of the study are not adequately addressed and there is almost no engagement with the issues surrounding the use of palaeopathological data and its limitations.

At present, there is very limited advancement of knowledge as many of the issues raised relate to information that is present in the cases that they have drawn together rather than new findings identified by the authors in the results. Where they have drawn new conclusions surrounding the lesion patterns, these are weak as they have not considered the biased nature of the sample they are using (already preselected for leprosy and the most severe forms at that) and the fact that they discarded cases that where not characteristic.

**Editorial and Data Presentation Modifications?**

Reviewer #1: See comments for authors attention in "Summary and General Comments" below.

Reviewer #2: Minor revisions

Reviewer #3: Minor comments

Abstract

The abstract does not make it clear why the research is important or needed, or moves beyond what is already known.

Introduction and methods

Just a note, Hansen’s Disease has its own problems since Hansen himself had some pretty unorthodox and unethical practices (https://worldneurologyonline.com/article/armauer-hansen-the-controversy-surrounding-his-unethical-human-to-human-leprosy-transmission-experiment/).

Page 3 Reference needed for incubation time

Reference for MB being the cause of most transmissions

Page 4 Leprosy was not eliminated from Europe by the 16th century, only parts of it. It was still found in Scandinavia, Iberia and some Eastern Areas for a long time after this.

How would changes in population density impact leprosy? Are you suggesting a link to the plague mortalities here?

In reference to citation 31. Is this 3-5% of modern patients?

P5 Criteria for inclusion and scoring. Where can we find the information about how these scores are defined. Why did you exclude Eurasian countries in Europe? What is your definition here, since Europe is a fairly modern construct. What about Belarus?

P6 – unless there are burial records we can not assign year of death, only a range.

In table headings, put references of the work that you are deriving your information from.

In data analysis, please let us know the program used for the tests, in addition please state whether you used correction factors for a) small sample sizes, and b) multiple testing (if appropriate).

Results

Make sure that before you present any %s you give the numbers observable.

Page 10, you don’t talk about the child cases that you discuss later. They are also not in the table. (page 12).

Data presentation. Please always let us know how many observations there were before providing any statistics (by individual or bone etc). This is critical so that we know how meaningful any percentages are. For example ‘Varying numbers of remains had data for specific cranial bones’ – very vague. In addition, please provide test statistic results (e.g. n values, test statistic and p Value).

Use of the term ‘ante mortem’. This could refer to any change on the skeleton before death, pathology, trauma, behavioural, activity related etc. I would say pathological changes so it is clear you are looking at disease-related change.

Page 13 You mention differences in cranial changes by age but you don’t say what you see in the data.

Nasal structures. Second sentence is incomplete?

P15 RMS – Tell us why you are referring to dental abscess data.

The data for the hands and feet is confusing. It would probably be more useful to present this by individual since there was likely high variability in the recovery of tarsals by individual. I know that Winchester had excellent recovery because they a) knew they had leprosy so explicitly focused on recovering hand and foot bones b) had properly train osteologists working on site. Whereas isolated cases in cemeteries might not have been looking for everything. As such, your numbers could be biased by differences in excavation practice.

P16 The increasing prevalence of inflammation (periostitis, osteomyelitis etc) with age is found in palaeopathology generally regardless of leprosy. Simply, the older you are, the more time you have for accumulating bone changes.

P18 – I am not sure about the title ‘ skeletons from modern day Europe’. Should this be skeletons from the region of modern day Europe?

Discussion

In the first paragraph of the discussion, the authors highlight the relevance of the topic to modern problems posed by the disease, but they don’t say how it can be used specifically. What has this specific piece of work contributed to these wider questions?

In terms of understanding M.leprae evolution, you do not discuss how the different strains are related to each other genetically and temporally. Also, I am not sure we can say which strains were prevalent for most geographic areas because for most places there are one or two cases. Only really for the UK or Denmark where multiple cases are analysed, and this would only be true for one period. There is no real new information presented here, rather just a repeat of what is known and presented directly in the genetic studies. For example Monot et al 2009, Mark 2017, Schuenemann et al 2018, Pfrengle et al 2021.

Page 23 – method for analysing the skulls from images. How many were done like this? Please put detail in the methods section for this. How reliable is this likely to be? Any intra observer tests for this?

P24 paragraph 3 ‘According to Moller Christensen, cranial bone changes associated with leprosy deformities in the hand and feet result in nasal alterations’ This does not make sense I am afraid. Rephrase.

You state that the proportion of RMS is higher in the 16th century, but is this due to them living longer, and the fact they are all from leprosaria (see above). Number of cases is also very low.

P25 Can you just outline why you are presenting the specific cases you are looking at them?

P25 – these cases are not described in your table previously, no children or babies are presented there. Were they excluded cases? This is the first time they are mentioned and as such the inclusion in the discussion is a bit random and doesn’t relate to any of the presented results.

In relation to the decline of leprosy, it is interesting that the lesion pattern is similar to S Korea, but a bit of caution is needed here, since a similar change could also be seen if there were improvements in care, or changes in care that allowed people to live longer with the disease. In addition, you did not find any clear statistical relationship between age and lesions. It is likely that the sample sizes are just too small to do this.

Top of page 27 In the UK, France, Spain and Belgium, ideas about the historic isolation of individuals with leprosy have been dismissed. Such isolation narratives have been particularly harmful to those with the disease today and are proven false. Please change the wording.

P28 – You can not use the high prevalence of nasal changes from a group of individuals that are identified because they have nasal changes as evidence for their importance in diagnosis. This is circular reasoning.

Yes, I totally agree that there is significant variation in the prevalence and type of bone alterations due to site selection, geography etc etc. Please can you discuss this in more detail.

People are undertaking NGS and WGS in archaeology and people are doing interdisciplinary papers. In particular there are multiple papers that combine genetics, geochemistry, pathology and dating. There are also a couple of papers looking at the role of non-human hosts (armadillos, squirrels). This latter is particularly important in modern contexts where animals are not considered as possible reservoirs for the disease.

**Summary and General Comments:**

Reviewer #1: LEPROSY IN SKELETONS FROM ARCHAEOLOGICAL SITES: A SYSTEMATIC REVIEW

Manuscript PNTD-D-24-01634.

Authors: Hugo Pessotti Aborghetti et al.

A). Leprosy main paper -review points.

The authors have reviewed a large number of studies (n=67) dealing with both the palaeopathology and biomolecular lineages of M. leprae cases isolated from antiquity to the post medieval period (3715 BCE to 1839 CE). A total of 297 skeletons with leprosy have been reviewed and the ontological data compiled and correlated by age, sex, period and various other categories.

The skeletal patterning due to leprosy has been much studied in the modern era since the late 1950’s, so osteological signs of damage due to M. leprae and secondary infections in the extremities of the appendicular skeleton and their associations with rhinomaxillary syndrome are well recognized. Consequently, many of the findings in the current review are not unexpected. The submission emphasizes the importance of combining palaepathological and biomolecular approaches leprosy but does not highlight specific areas for future study.

It is appreciated that this undertaking represents a great deal of data mining and assessment from a large body of work. The review is very comprehensive but does not mention one of the rarer manifestations of multibacillary leprosy sometimes seen in adolescent remains, leprogenic odontodysplasia (LO) (Danielsen K. Tandlaegebladet 1970; 74:603–625). The phenomena has only been described in the palaeopathological literature and invariably from North-Western Europe. The current submission cites this in included reference article 39 (Matos & Santos. 2013) and in the main article (ref 46) but does not record detail. LO has also been seen in 4 individuals from the St. Mary Magdalen leprosaria in Winchester, Hants. UK (article ID 66 and main text ref. 59). For completion, the appropriate place to record this would be the Data Extraction Table alongside the other dental pathology.

Genomic studies.

1. On p20 the authors state that 5 previous papers have looked for evidence of M. lepromatosis in genomic studies. In fact, an additional group looked and this is detailed in your main text reference 106, Mendum et al, 2018.

2. SNP type 3. This genotype is mentioned a few times within the main text. E.g. Abstract, p20, p23, This category has come to be recognized as a broad group within the Monot typing scheme, which originally described strains as belonging to SNP types 1A to 4P. Type 3 includes the oldest lineage 3K, itself sub-divided into 3K0 and 3K1, as well as 3M, now seen as nearer/basal to SNP-type 4 strains and type 3I-1, responsible for the majority of European cases in the mediaeval period and an ancestor to the 31-1 and 3I-2 types present in southern USA states (Truman et al., N. Engl. J. Med. 364, 1626, 2011). This reviewer suggests that where known, the subtype is provided in the text to provide better context; predominantly this is likely to be 3I. Similarly, the most commonly identified SNP-type 2 strain found has been 2F.

In the concluding remarks on p29, the authors mention the importance of integrating genomic studies into future research. This is undoubtedly true, but broader biomolecular approaches may also grow in importance and provide alternative means of studying the mycobacterium. For example, mycolipids, proteins and peptides from M. leprae.

Minor points.

1. Leprosaria (found on p2, p3, p10, p26, p27, p28) is usually italicized by convention to denote its Latin origins.

2. “Leprosy presents a spectrum of skin and nerve clinical presentations”. This could be rephrased to better convey the intended meaning with corrected grammar.

B). Supplementary Information (Appendices). xlsx file.

The following points need attention.

1. Data extraction tab, columns BT & BU. Spelling of Skulls.

2. Genomic References included Tab. “Synopsis”. Entry 3. The 2G genotyping of Economou’s paper was later corrected to 2F, in line with findings from other groups. This correction is published in JAS 40 (6), page 2867.

3. QA Ratings (genomic) Tab. Entry 23/24, Roffey 2017. It is surprising to see this assessed as 00 for biomolecular analysis when this reference successfully typed 9 SNP and 3 VNTR loci to show this was a type 2F. DNA preservation was later shown to be good enough for WGS which confirmed this genotype. Details in reference 60. Perhaps this Roffey MS has been confused with Roffey, 2012 ?

4. Included References (Genomic) Tab. “Synopses” missing from Monot et al, 2009 and Taylor et al, 2018. Also, type 3I-1 found in Entry 21 (see Main text comment above).

5. QA Ratings Tab, Item 6 Biomolecular Analysis. Many of the publications here do not offer genomic analysis yet have been assessed as 00 by the two independent reviewers. It would be clearer if these are flagged as either ND (not determined) or NA (not available). These publications include Blau, 2005; Baker 2014; Buckley 2008 etc.

6. Data Extraction Tab. Article ID 11 (Cole, 2022) of Appendices shows ? for total skeletons at the site. Having checked with one of the excavators, I can offer that the skeleton total excavated and analyzed to date is 125, as mentioned in Discussion in J. Med. Microbiol. 2024; 73: 001806.

Reviewer #2: Summary: This systematic review presents a comprehensive analysis of leprosy in archaeological skeletons, synthesizing data on skeletal changes and genomic findings to explore the historical trajectory of the disease. The study provides valuable insights into the paleopathology of leprosy, including the frequency and distribution of Rhinomaxillary Syndrome (RMS) and post-cranial bone changes (PCBC). The manuscript is well-structured, methodologically sound, and contributes significantly to our understanding of leprosy’s historical impact. With minor revisions incorporating a One Health perspective and a more nuanced discussion of the decline hypothesis, the paper will be well-suited for publication. The revisions will enhance the interdisciplinary relevance of the study and align it with contemporary approaches to infectious disease history and epidemiology.

Strengths:

1. Comprehensive Scope: The systematic review effectively synthesizes data from multiple archaeological studies across different regions and time periods.

2. Robust Methodology: The inclusion criteria, data extraction, and quality assessment procedures are clearly defined and rigorously applied.

3. Genetic Analysis Integration: The discussion on Mycobacterium leprae genotypes strengthens the argument regarding leprosy’s historical spread and decline.

4. Clarity of Presentation: The figures and tables are well-organized, enhancing the readability and interpretability of results.

Proposed revisions:

1. Incorporating a One Health Perspective:

o The discussion on the decline of leprosy in medieval Europe primarily focuses on immunological, epidemiological, and environmental factors. However, it lacks consideration of the broader One Health framework, which integrates human, animal, and environmental health interactions.

o Suggested Revision: Expand the discussion to consider how zoonotic reservoirs, changes in human-animal interactions, and environmental shifts may have influenced the disease’s decline. For instance, studies suggest that other mycobacterial species present in the environment may have played a role in cross-immunity.

2. Clarification of the Decline Hypothesis:

o The current hypothesis emphasizes tuberculosis-leprosy co-infection, climate change, and improved living conditions as key factors in the decline of leprosy.

o Suggested Revision: While these factors are valid, it would be beneficial to explore the role of urbanization, changing human demographics, and potential shifts in animal host populations that could have influenced transmission dynamics.

3. Contextualizing the Findings with Modern Epidemiology:

o The discussion draws valuable parallels between historical and modern leprosy cases. However, additional references to contemporary One Health approaches in leprosy control would strengthen this connection.

o Suggested Revision: Briefly mention current One Health initiatives that address zoonotic transmission of leprosy (e.g., armadillos in the Americas as reservoirs of M. leprae) and how similar dynamics might have played a role in historical contexts.

4. Minor Edits for Clarity and Consistency:

o Ensure consistency in the use of terminology, particularly in references to ‘possible’ and ‘probable’ RMS.

o Some sections could benefit from minor grammatical improvements for readability.

o This also applies to the styling of the references.

5. Selection of references is only partly accounted for:

There is a discrepancy between the 86 in the systematic review and the overall 134 references. 40% of the references are not accounted for.

Reviewer #3: This paper aims to bring together all of the evidence from published leprosy cases from across the globe with a view to better understanding its evolutionary history, temporal geographic spread and changes in lesion presentation. It does this through a literature review and some reassessment of images. The paper is generally well written and within the scope of interest of the journal readership. That being said, I have some major reservations about the work that need addressing before it can be accepted for publication. I hope that these comments are useful for the authors so that they make get their work published.

A major problem with this paper is that there is a lack of critical thought about the palaeopathological data, its limitations and meaning. While I think what the authors suggest from the data is probably correct, there are multiple explanations for what they have observed and these should be addressed. In addition, there is no specific research question(s), or aims which means the results and discussion is a bit haphazard and unfocused. A hypothesis or two would help here.

A significant limitation is that the sample is pre-selected for having leprosy. That is, the lesions were usually distinct enough that someone was confident that the individual could be diagnosed with leprosy in the first place. In sticking only to cases that they can be confident in recognising then we are really only looking at the most severe manifestation of the disease. Given we miss the cases of leprosy without skeletal lesions, or discard the ones with less obvious lesions, it is very difficult to compare what we have with modern data generally. Furthermore, when talking about disease presence/prevalence, we also have to be very careful (discussion page 22). The presence of leprosaria in particular are problematic because they may be acting to concentrate cases giving a perception that the disease was becoming increasingly common. To understand this, we would need to know far more about leprosaria and what their ‘catchments’ may have been as well as more precise dating of burials - this does not exist yet. This would then need to be considered with the context of increasing population size through time. Yes, there were more cases in the late medieval period, but the population was also much larger than it had been before!

This preselection for diagnostic leprosy cases is also problematic for understanding lesion patterns too. Those lesions that are diagnostic for leprosy are going to be inflated in comparison to other types. As such, I am not surprised that RMS was observed in 80% of cases given this is likely the clue that got them accepted as a case and then published! Additionally, the increase in RMS over time in England could well be because those in leprosaria were getting support so lived longer and developed more pronounced lesions. At Winchester, special feeding equipment was found. Furthermore, for the less diagnostic and non-specific lesions, we need to know how common they are they in the populations in general, and what can be attributed to the disease. What would be more interesting would be an aged-matched comparison of lesions from individuals in leprosaria with those outside leprosaria as this may tell us about efficacy of treatment and life outcomes of people with the condition.

This paper needs to discuss the current issues with inter and intraobserver error that exists between palaeopathologists. There is no detailed methodology for recording post cranial changes at present so what someone might score as present, another might not. This has been the significant factor that has really prevented such a large-scale study such as that presented here. It is simply very difficult to reliably compare data. How can you overcome this problem?

In many parts of the discussion, the authors draw attention to a number of issues or themes surrounding leprosy evolution/history yet they have not linked this to new data that they have produced. For example, the link between TB and leprosy and the decline of the latter, or the distributions of the strains. They need to explicitly show how their data adds new information to these discussions beyond what is already discussed in the references and citations, or remove it from the text.

PLOS authors have the option to publish the peer review history of their article (what does this mean? ). If published, this will include your full peer review and any attached files.

**Do you want your identity to be public for this peer review?** For information about this choice, including consent withdrawal, please see our Privacy Policy .

Reviewer #1: No

Reviewer #2: **Yes: ** Toine Pieters

Reviewer #3: No

**Figure resubmission:**
---

## [Decision Letter · Decision Letter 1]

10 Jul 2025

PNTD-D-24-01634R1LEPROSY IN SKELETONS FROM ARCHAEOLOGICAL SITES: A SYSTEMATIC REVIEWPLOS Neglected Tropical Diseases

Dear Dr. Deps,

Thank you for submitting your revised manuscript to PLOS Neglected Tropical Diseases. After careful consideration, we feel that it has merit but does not fully meet PLOS Neglected Tropical Diseases's publication criteria as it currently stands. Therefore, we invite you to submit a revised version of the manuscript that addresses the points raised during the review process. Please submit your revised manuscript within 30 days Aug 09 2025 11:59PM. If you will need more time than this to complete your revisions, please reply to this message or contact the journal office at plosntds@plos.org. When you're ready to submit your revision, log on to https://www.editorialmanager.com/pntd/ and select the 'Submissions Needing Revision' folder to locate your manuscript file.

* A rebuttal letter that responds to each point raised by the editor and reviewer(s). You should upload this letter as a separate file labeled 'Response to Reviewers '. This file does not need to include responses to any formatting updates and technical items listed in the 'Journal Requirements' section below.* A marked-up copy of your manuscript that highlights changes made to the original version. You should upload this as a separate file labeled 'Revised Manuscript with Track Changes '.* An unmarked version of your revised paper without tracked changes. You should upload this as a separate file labeled 'Manuscript '. If you would like to make changes to your financial disclosure, competing interests statement, or data availability statement, please make these updates within the submission form at the time of resubmission. Guidelines for resubmitting your figure files are available below the reviewer comments at the end of this letter.

We look forward to receiving your revised manuscript.

Kind regards, Katharina RöltgenAcademic EditorPLOS Neglected Tropical Diseases Mathieu PicardeauSection EditorPLOS Neglected Tropical Diseases

Shaden Kamhawi

co-Editor-in-Chief

Paul Brindley

co-Editor-in-Chief

**Journal Requirements:** 1) We have noticed that you have uploaded Supporting Information files, but you have not included a complete list of legends. Please add a full list of legends for your Supporting Information file "PRISMA_2020_checklist.docx " after the references list.  **Reviewers' comments:** Reviewer's Responses to Questions

**Key Review Criteria Required for Acceptance?**

**Methods**

-Are the objectives of the study clearly articulated with a clear testable hypothesis stated?

-Is the study design appropriate to address the stated objectives?

-Is the population clearly described and appropriate for the hypothesis being tested?

-Is the sample size sufficient to ensure adequate power to address the hypothesis being tested?

-Were correct statistical analysis used to support conclusions?

-Are there concerns about ethical or regulatory requirements being met?

Reviewer #1: (No Response)

Reviewer #2: The methods are adequately described

Reviewer #3: This work is much improved and I thank the authors for taking my comments on board.

I would suggest that they are clearer about whether the lesions scored for the post crania can be directly connected to leprosy or not. over 50% of femurs being affected is high, and importantly much higher than the tibia or fibula. This is highly irregular as the femur is rarely affected. Is this a typo?

**Results**

-Does the analysis presented match the analysis plan?

-Are the results clearly and completely presented?

-Are the figures (Tables, Images) of sufficient quality for clarity?

Reviewer #1: (No Response)

Reviewer #2: There is a mistake in the closure date of the Batavia leprosarium in the table: it should be 1896

The results are clearly and completely presented.

Reviewer #3: (No Response)

**Conclusions**

-Are the conclusions supported by the data presented?

-Are the limitations of analysis clearly described?

-Do the authors discuss how these data can be helpful to advance our understanding of the topic under study?

-Is public health relevance addressed?

Reviewer #1: (No Response)

Reviewer #2: I have two remarks with regard to the conclusions:

1. Try to include the reults of the follwing study: Urban C, Blom AA, Avanzi C, Walker-Meikle K, Warren AK, White-Iribhogbe K, Turle R, Marter P, Dawson-Hobbis H, Roffey S, Inskip SA, Schuenemann VJ. Ancient Mycobacterium leprae genome reveals medieval English red squirrels as animal leprosy host. Curr Biol. 2024 May 20;34(10):2221-2230.e8. doi: 10.1016/j.cub.2024.04.006.

2. In the twentieth century there still is endemic leprosy in Spain and Portugal: See: Suárez-García I, Gómez-Barroso D, Fine PEM. Autochthonous leprosy in Spain: Has the transmission of Mycobacterium leprae stopped? PLoS Negl Trop Dis. 2020 Sep 16;14(9):e0008611. doi: 10.1371/journal.pntd.0008611.

Ferreira PM, Rato IR, Rigor J, Mota M. Hansen's disease - a forgotten disease? JRSM Open. 2021 Aug 30;12(8):20542704211035995. doi: 10.1177/20542704211035995.

Reviewer #3: (No Response)

**Editorial and Data Presentation Modifications?**

Reviewer #1: (No Response)

Reviewer #2: No modifications needed.

Reviewer #3: (No Response)

**Summary and General Comments**

Reviewer #1: I consider the authors have now revised their manuscript, taking into account the points raised in review or reasonably rebutted individual issues.

Reviewer #2: Okay apart from the false statement that leprosy in Europe was eliminated in the 19th century.

Reviewer #3: (No Response)

PLOS authors have the option to publish the peer review history of their article (what does this mean? ). If published, this will include your full peer review and any attached files.

**Do you want your identity to be public for this peer review?** For information about this choice, including consent withdrawal, please see our Privacy Policy .

Reviewer #1: No

Reviewer #2: No

Reviewer #3: No

---

## [Editor Report · Decision Letter 2]

17 Jul 2025

Dear Professor Deps,

We are pleased to inform you that your manuscript 'LEPROSY IN SKELETONS FROM ARCHAEOLOGICAL SITES: A SYSTEMATIC REVIEW' has been provisionally accepted for publication in PLOS Neglected Tropical Diseases.

Best regards,

Katharina Röltgen

Academic Editor

Mathieu Picardeau

Section Editor

Shaden Kamhawi

co-Editor-in-Chief

Paul Brindley

co-Editor-in-Chief

---

## [Editor Report · Acceptance letter]

Dear Professor Deps,

We are delighted to inform you that your manuscript, "LEPROSY IN SKELETONS FROM ARCHAEOLOGICAL SITES: A SYSTEMATIC REVIEW," has been formally accepted for publication in PLOS Neglected Tropical Diseases.

Best regards,

Shaden Kamhawi

co-Editor-in-Chief

Paul Brindley

co-Editor-in-Chief
